# Glycan remodeled erythrocytes facilitate antigenic characterization of recent A/H3N2 influenza viruses

Frederik Broszeit[1,6], Rosanne J. van Beek[1,6], Luca Unione[1], Theo M. Bestebroer[2], Digantkumar Chapla[3], Jeong-Yeh Yang[3], Kelley W. Moremen [3], Sander Herfst [2], Ron A. M. Fouchier [2], Robert P. de Vries[1✉] & Geert-Jan Boons [1,3,4,5✉]

During circulation in humans and natural selection to escape antibody recognition for decades, A/H3N2 influenza viruses emerged with altered receptor specificities. These viruses lost the ability to agglutinate erythrocytes critical for antigenic characterization and give low yields and acquire adaptive mutations when cultured in eggs and cells, contributing to recent vaccine challenges. Examination of receptor specificities of A/H3N2 viruses reveals that recent viruses compensated for decreased binding of the prototypic human receptor by recognizing α2,6-sialosides on extended LacNAc moieties. Erythrocyte glycomics shows an absence of extended glycans providing a rationale for lack of agglutination by recent A/H3N2 viruses. A glycan remodeling approach installing functional receptors on erythrocytes, allows antigenic characterization of recent A/H3N2 viruses confirming the cocirculation of antigenically different viruses in humans. Computational analysis of HAs in complex with sialosides having extended LacNAc moieties reveals that mutations distal to the RBD reoriented the Y159 side chain resulting in an extended receptor binding site.

[1] Department of Chemical Biology & Drug Discovery, Utrecht Institute for Pharmaceutical Sciences, Utrecht University, Utrecht 3584 CG, The Netherlands. [2] Department of Viroscience, Erasmus MC, P.O. Box 2040, Rotterdam 3000 CA, The Netherlands. [3] Complex Carbohydrate Research Center, University of Georgia, 315 Riverbend Rd, Athens, GA 30602, USA. [4] Bijvoet Center for Biomolecular Research, Utrecht University, Utrecht, The Netherlands. [5] Department of Chemistry, University of Georgia, Athens, GA 30602, USA. [6]These authors contributed equally: Frederik Broszeit, Rosanne J. van Beek. ✉email: r.vries@uu.nl; g.j.p.h.boons@uu.nl

Human influenza A viruses have a remarkable ability to evolve and evade neutralization by antibodies elicited by prior infections or vaccinations[1,2]. This antigenic evolution, or drift, is mainly caused by amino acid substitutions in the globular head of the hemagglutinin (HA) protein where binding occurs with sialic acid receptors of host cells[3,4]. These substitutions in circulating influenza viruses lead to antigenic differences as compared to employed vaccines, resulting in poor vaccine-mediated protection[5]. Therefore, the World Health Organization (WHO) Global Influenza Surveillance and Response System (GISRS) continuously monitors antigenic changes in circulating influenza viruses and recommends updated compositions of influenza vaccines biannually[6].

Antigenic surveillance and vaccine strain selection rely predominantly on the hemagglutination inhibition (HI) assay, in which the ability of serum antibodies to block receptor binding by the influenza virus HA protein is quantified[7]. Such antibodies prevent virus-mediated agglutination of erythrocytes and are measured as a correlate of protection[8]. The HI assay makes it possible to select virus strains that are antigenically representative of circulating viruses for vaccine development[9–11]. It is easy to perform in a high throughput manner, can be standardized and is highly reproducible across laboratoria[12].

A/H3N2 viruses, which have become the leading cause of seasonal influenza illness and death since their introduction in the human population in 1968[13,14], exhibit a particularly rapid antigenic drift. As a result, the WHO has recommended 28 vaccine strain updates since these viruses started circulating in humans[13]. The rapid antigenic evolution of A/H3N2 viruses coincided with altered receptor usage, which in turn has resulted in an inability to agglutinate erythrocytes commonly employed for HI assays. As a result, antigenic characterization of circulating A/H3N2 viruses using the HI assay is increasingly difficult, complicating the selection of appropriate vaccine strains[1,8–10]. The receptor-binding phenotype of recent A/H3N2 viruses is also hampering virus replication under laboratory conditions for amplification of clinical isolates and leads to adaptive substitutions when grown in embryonated chicken eggs[15,16]. The difficulties to antigenically characterize circulating A/H3N2 viruses, in particular those belonging to the dominant 3C.2a clade, and the inability of large-scale virus production without egg-adaptation has led to serious problems with A/H3N2 influenza vaccine production and effectiveness[1,11].

HAs of human influenza viruses bind to cell surface glycans carrying terminal α2,6-linked sialic acid moieties (Neu5Ac(α2,6)Gal), which is referred to as the prototypic human receptor[17]. These receptors are usually part of N-linked glycans, which are highly complex biomolecules composed of a core pentasaccharide modified by various numbers and patterns of branching N-acetylglucosamine (GlcNAc) moieties[18]. These branching points can be extended by several N-acetyl-lactosamine (Gal(β1,4)GlcNAc, LacNAc) repeating units, which in turn can be capped by various types of fucosylation and sialylation. Several studies have shown that not only the α2,6-linked sialoside but also the underlying oligosaccharide structure of N-linked glycans can contribute to HA binding selectivities[19–23]. Antigenic pressure results mainly in amino acid substitutions near the receptor-binding site of HA, which may result in changes in glycan receptor specificities. Thus, failure of contemporary A/H3N2 viruses to agglutinate erythrocytes, the poor replication in mammalian cells and the emergence of egg adaptive mutations are probably due to an adaptation to glycan receptors that are not expressed by these cell substrates. Understanding of the evolution of receptor usage by A/H3N2 viruses at a molecular level will open avenues to address challenges in surveillance and vaccine production for these viruses.

In this work, we examined receptor specificities of A/H3N2 viruses representing different evolutionary time points and clades using a novel glycan microarray to identify the minimal receptor requirements for recent, non-agglutinating A/H3N2 strains. Glycan analysis of cell surface N-glycans on various erythrocytes reveal an absence of such receptors. We developed an exo-enzymatic cell surface glycan remodeling strategy to install appropriate receptors on fowl erythrocytes to regain binding by contemporary A/H3N2 viruses. Such viruses can agglutinate the glyco-engineered erythrocytes and made it possible to antigenically characterize A/H3N2 viruses by HI assays. We reveal substantial antigenic differences of circulating virus isolates to current vaccine strains providing a rationale for poor vaccine performance.

## Results

**Glycan microarray analysis to determine A/H3N2 receptor specificity.** Although glycan microarray technology has been used to examine receptor requirements of HAs[24], these were not populated with biologically relevant glycans to establish minimal receptor requirements. This information is, however, critical to understand how receptor binding has evolved over time and how a lack of expression of specific glycans by erythrocytes or laboratory hosts may have resulted in a loss of agglutination or a lack of propagation, respectively.

We have constructed a glycan array that is populated with biologically relevant bi-antennary N-glycans having different numbers of LacNAc repeating units in various structural configurations. They resemble structures found on human respiratory tissue, which abundantly expresses N-glycans having multiple consecutive LacNAc repeating units that can be capped by sialic acid[25,26]. The synthetic glycans are either unmodified (compounds **1**-**3**), capped by avian α2,3-linked (compounds **4**-**6**), or human α2,6-linked sialosides (compounds **7**-**17**). Most naturally occurring N-linked glycans have asymmetrical architectures in which the various antennae are modified by oligo-LacNAc moieties of different lengths[27]. To probe the importance of such architectures for HA recognition, we prepared symmetrical as well as asymmetrical glycans (**8**, **9**, **11**, **12**, **13**, **15**, **16**, and **17**) that are modified by either one or two sialosides linked to LacNAc chains of different length. The collections of compounds made it possible to probe the importance of mono- vs. bidentate binding interactions, and a possible preference or requirement for a sialoside at specific antenna or at an extended LacNAc chain. All glycans contain an anomeric asparagine moiety and its α-amine facilitated immobilization on amine reactive, NHS-activated glass slides. The quality of the printing was validated by probing the array with the lectins ECA, SNA, and MAL1 as well as an anti-H3 antibody (Fig. 1a and Supplementary Fig. 1).

The glycan array was probed with various A/H3N2 viruses, representing distinct evolutionary time points and clades and having different abilities to agglutinate erythrocytes derived from species commonly used in HI assays (Supplementary Fig. 2). In this respect, chicken and turkey erythrocytes are widely employed in HA assays because they are nucleated, which unlike mammalian erythrocytes, sediment rapidly thereby greatly facilitating the visual readout of the assay[7]. They express α2,3- and α2,6-linked sialosides and can be agglutinated by avian as well as human influenza viruses. Other types of erythrocytes have been employed for HI assays[28] and in particular those of guinea pig erythrocytes have been proven to be useful because they exhibit a somewhat broader agglutination ability and can for example be employed to antigenically characterize H3N2 viruses of the 3C.2 clade which cannot be agglutinated turkey and chicken erythrocytes[29].

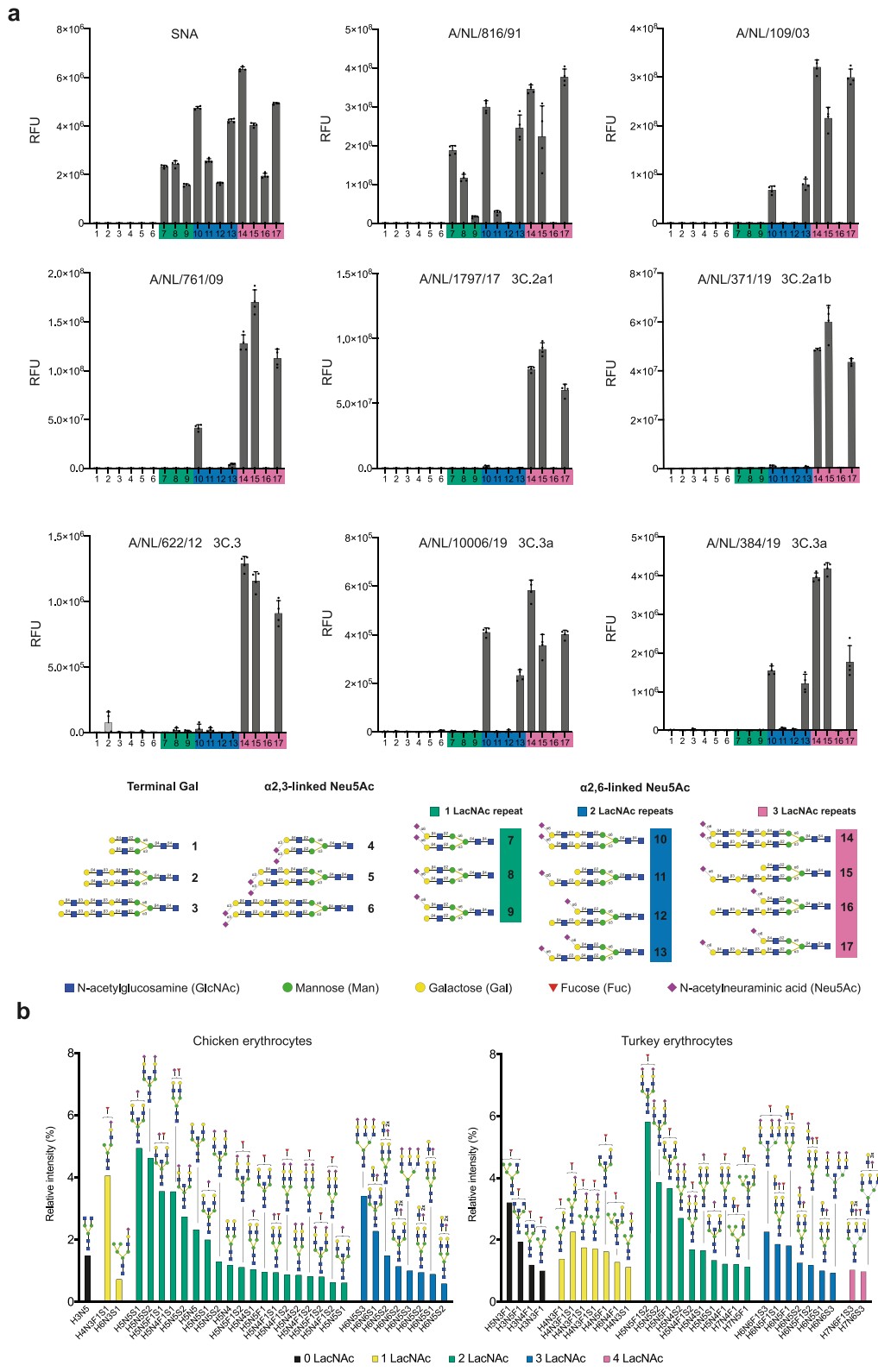

**Fig. 1 Receptor-binding specificities of representative A/H3N2 viruses using glycan microarray analysis and glycomic analysis of *N*-glycosylation of unmodified fowl erythrocytes. a** Binding was visualized using a human anti-H3 stalk antibody (CR8020). Bars represent the background-subtracted average relative fluorescence units (RFU) of four replicates ±SD. Values for all individual datapoints are represented in the Supplementary Source Data file. **b** The 30 most abundant *N*-glycans on erythrocytes from chicken and turkey organized by the number of LacNAc repeating units and relative intensity. High-mannose type *N*-glycans are not shown and their abundance is presented in Supplementary Fig. 5. The structures of all detected glycans are shown in Data S1.

In the studies, we included A/NL/816/91 (NL91) which can agglutinate chicken, turkey, and guinea pig erythrocytes, A/NL/109/03 (NL03) which only agglutinates turkey and guinea pig erythrocytes[29] and A/NL/761/09 (NL09) which only agglutinates α2,6-resialylated turkey and guinea pig erythrocytes[30–32]. During the past decade, A/H3N2 viruses have evolved into distinct, cocirculating antigenic groups, referred to as clades (Supplementary Fig. 2). We examined A/NL/1797/17 (NL17) and A/NL/371/19 (NL19) as recent examples of the 3C.2a clade that insufficiently hemagglutinate all commonly used erythrocytes for HI assays, and poorly infect MDCK cells[33]. The 3C.3a clade is represented by A/NL/10006/19 and A/NL/384/19, and their evolutionary predecessor A/NL/622/12 (3C.3). Although 3C.3 viruses cannot agglutinate any erythrocyte type, 3C.3a viruses are unique as they regained an ability to agglutinate turkey and guinea pig erythrocytes (Supplementary Fig. 2).

Whole viruses were applied to the microarray and detection of binding was accomplished by a human anti-H3 stalk antibody (CR8020) (Supplementary Fig. 1). NL91 recognized most of the human-type receptors, including compounds that have an α2,6-sialoside on a mono-LacNAc residue (glycans **7**-**9**, Fig. 1a and Supplementary Fig. 3). Compound **8** exhibited a substantial greater responsiveness compared to **9** indicating that this virus has a preference for a sialoside at the α1,3-arm. Interestingly, the sialyltransferase, ST6Gal1, which is solely responsible for installing human-type receptors, preferentially modifies the α1,3-arm of N-linked glycans[34]. Compounds **7** and **8** did bind similarly demonstrating that an additional sialic acid at the α1,6-arm does not substantially contribute to binding. Another unanticipated observation was that compounds **12** and **16** did not exhibit binding whereas **9** showed responsiveness highlighting that an extended and unmodified LacNAc moiety at the α1,3-arm can block recognition of the other arm. Collectively, the results show that the minimal receptor for NL91 is a bi-antennary N-glycan having two LacNAc moieties modified by a single sialoside (glycan **8**).

NL03 and NL09 recognized far fewer glycans and did not bind to structures having their α2,6-sialosides at a mono-LacNAc moiety (**7**-**9**, **12** and **16**). This observation indicates that the minimal receptor for these viruses is a bis-sialylated N-glycan having at least one di-LacNAc moiety (glycan **13**). NL17 and NL19 (3C.2a) showed only strong responsiveness to **14**, **15**, and **17**. These glycans have in common that at least one of the arms is extended by three consecutive LacNAc units that is further modified by an α2,6-sialoside. Thus, a glycan having four LacNAc units arranged in an asymmetrical manner (**15**) represents the minimal receptor for these viruses. Mono-sialylated derivative **15** gave a similar responsiveness compared to the bis-sialosides **14** and **17** indicating that a bidentate binding event does not substantially contribute to recognition as previously suggested[22]. Instead, it appears that reduction in binding of an α2,6-sialyl-Gal moiety, which is widely regarded as the prototypic human receptor, has been compensated by recognition of sialosides at extended LacNAc chains. 3C.3a viruses (A/NL/10006/19 and A/NL/384/19) exhibited a similar binding profile as NL03 and 09 and bound to bis-sialosides **10** and **13** having the Neu5Ac residue at a di-LacNAc chain. Interestingly, their ancestor (A/NL/622/12, 3C.3) required the Neu5Ac residue to be presented on a tri-LacNAc structure similar to the requirement of 3C.2a viruses. Thus, recent 3C.3a viruses have regained an ability to recognize shorter structures.

**Glycomic analysis of chicken and turkey erythrocytes**. Next, we examined structures of N-linked glycans expressed by chicken and turkey erythrocytes and compared the data with the receptor requirements of the various A/H3N2 viruses. Membrane fractions of the cells were treated with PNGase F to release the N-glycans which were isolated by solid phase extraction using C18 and Porous Graphitized Carbon (PGC) cartridges, and then analyzed by liquid chromatography mass spectrometry (LC-MS)[35]. The 30 most abundant complex type N-glycan compositions for the two cell types are presented in Fig. 1b. The compounds are organized according to an increasing number of LacNAc moieties (indicated by different color coding) and include hybrid-type and core structures having none (black bars) or 1 LacNAc moiety (yellow bars) and complex N-glycans having variable numbers of LacNAc repeating units (2–4 LacNAc, green, blue and purple bars, respectively). High-mannose type N-glycans were also detected and their structures and abundance are shown in Supplementary Fig. 5. Strikingly, chicken erythrocytes do not substantially express N-glycans having 4 LacNAc units which is the minimal epitope requirement for contemporary non-agglutinating A/H3N2 viruses. Turkey erythrocytes do express some glycans with this number of LacNAc units (purple bars in Fig. 1b), but the majority was assigned as tri- and tetra-antennary glycans because of substitution with three or four sialic acids. The latter was supported by selective release of bi-antennary N-glycans by Endo F2, and in this case LC-MS analysis did not detect glycans having four LacNAc moieties (Supplementary Fig. 4). Thus, turkey erythrocytes also do not substantially display sialylated epitopes having three consecutive LacNAc moieties. The majority of the glycans released by Endo F2 lacked fucose indicating that the fucosides observed in some of the structures depicted in Fig. 1 can be assigned to core modification. Chicken erythrocytes express substantial quantities of high mannose glycans (Supplementary Fig. 5) whereas turkey cells display almost none of these structures. The greater abundance of complex type glycans on turkey erythrocytes offers a possible rationale for the ability of the NL03 and NL09 A/H3N2 viruses to agglutinate unmodified or α2,6-resialylated turkey erythrocytes, respectively.

**Erythrocyte glycoengineering to install functional receptors**. We embarked on a strategy to enzymatically remodel glycans of fowl erythrocytes to install receptors for A/H3N2 viruses of the 3C.2 clade to make them suitable for HI assays (Fig. 2a). Treatment of erythrocytes with a neuraminidase was expected to remove sialic acids and reveal terminal galactosides which are appropriate acceptors for installing additional LacNAc moieties. The latter residues can be introduced by the concerted action of the enzymes B4GalT1 and B3GnT2, which sequentially install β1,4-linked galactoside and β1,3-linked N-acetyl-glucosamines, respectively. The terminal galactosides of the resulting extended LacNAc moieties can then be modified by the sialyltransferase ST6Gal1 to install terminal α2,6-linked sialosides[36]. The enzymatic remodeling was conveniently performed by incubating the erythrocytes with the neuraminidase from *Arthrobacter ureafaciens* for 6 h after which B4GalT1, B3GnT2[37], UDP-Gal, and UDP-GlcNAc were added followed by incubation overnight. Next, the cells were pelleted by centrifugation, washed to remove the enzymes and sugar nucleotides, and then incubated with ST6Gal1 in the presence of CMP-Neu5Ac for 4 h. Glycomic analysis of the resulting cells, which were denoted as 2,6-Sia Poly-LN cells, confirmed that the antennae of the N-linked glycans had been extended by additional LacNAc moieties, and both cell types expressed sialylated structures having four LacNAc units (Fig. 2b, 4 LacNAc units are indicated in purple bars). Analysis of glycans on turkey erythrocytes released by Endo F2 treatment confirmed the presence of bi-antennary glycans that have 4 LacNAc moieties and are potentially suitable receptors for contemporary A/H3N2 viruses (Supplementary Fig. 4, glycans having 4 LacNAc units are

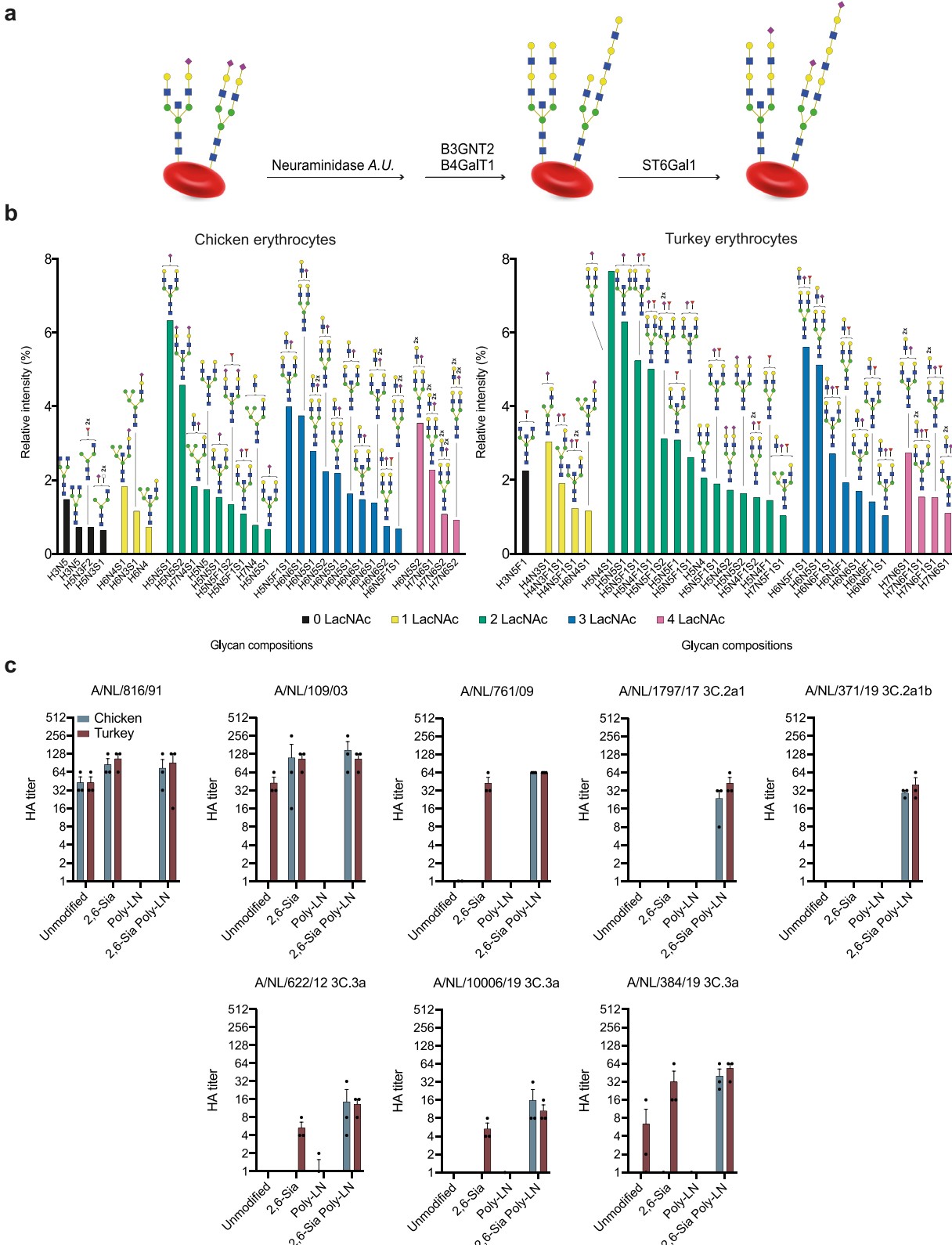

indicated by purple bars). Chicken and turkey erythrocytes express a mixture of α2,3- and α2,6-linked sialosides. To examine whether an increase in the abundance of α2,6-sialosides would improve agglutination, control cells were prepared by treatment with neuraminidase and resialylation with ST6Gal1 (denoted as 2,6-Sia cells). As negative control, we employed cells that have extended LacNAc moieties but lack sialic acids (denoted as Poly-LN cells).

Phenotypic properties of the glyco-engineered erythrocytes were examined using the hemagglutination (HA) assay (Fig. 2c). As expected, NL91 agglutinated unmodified, 2,6-Sia and 2,6-Sia poly-LN erythrocytes, which was in agreement with the finding that these viruses can employ N-glycans that have simple and extended α2,6-sialylated structures. NL03 agglutinated unmodified turkey erythrocytes, but interestingly also α2,6-resialylated

**Fig. 2 Schematic overview of enzymatic modification of erythrocytes followed by glycomic analysis of *N*-glycosylation and their use in hemagglutination assays. a** Neuraminidase *A.U.*: Neuraminidase from *Arthrobacter ureafaciens*; B3GnT2: β-1,3-*N*-acetylglucosaminyltransferase 2; B4GalT1: β-1,4-galactosyltransferase 1; ST6Gal1: α-2,6-sialyltransferase 1. **b** The 30 most abundant *N*-glycans on the enzymatically modified erythrocytes from chicken and turkey (based on relative intensity, excluding high-mannose type *N*-glycans, for all structures refer to Data S2) sorted by abundance and number of LacNAc units. Proposed structures are assigned to detected glycan compositions. **c** A/NL/816/91, A/NL/109/03, A/NL/761/09, A/NL/1797/17, A/NL/371/19, A/NL/622/12, A/NL/10006/19, and A/NL/384/19 tested with modified erythrocytes (2,6-Sia Poly-LN) from chicken (blue) and turkey (red). Unmodified, 2,6 resialylated (2,6-Sia) and extended desialylated (Poly-LN) erythrocytes were added as controls. Assays were performed in biological triplicates in the presence of oseltamivir and the means ± SEM were plotte.

chicken erythrocytes. The latter may be due to an increase in the abundance of α2,6-linked sialosides having two consecutive LacNAc repeatin units, which are present on chicken erythrocytes. α2,6-Resialylation of turkey erythrocytes was sufficient to recover agglutination of NL09, and in this case the greater abundance of α2,6-sialylation on extended structures already present on these cells is probably responsible for the improved agglutination. Importantly, NL17 and NL19 (3C.2a) agglutinated only erythrocytes that were enzymatically remodeled to have extended sialylated LacNAc moieties (2,6-Sia Poly-LN cells). Similar results were obtained for A/NL/622/12 (3C.3), which is in agreement with receptor requirements similar to viruses of the 3C.2a clade. As anticipated, the 3C.3a viruses (A/NL/10006/19 and A/NL/384/19), which have reverted to recognize shorter structures, could also agglutinate α2,6-resialyated turkey erythrocytes.

Next, HA assays were performed with a wider collection of A/H3N2 viruses (Table 1) to validate the robustness of the glycoengineering method with a focus on contemporary A/H3N2 viruses that have lost the ability to hemagglutinate unmodified erythrocytes and do not replicate efficiently in wild-type MDCK cells. Several recent vaccine strains heavily adapted to growing in eggs (X-161B, IVR-147, X-223A, NIB-104, NIB-112, and X-327), an A/H1N1 (A/Singapore/GP1908/15) and an influenza B (B/Maryland/15/15) strain were included as controls. As expected, pre-2000 A/H3N2 (A/Bilthoven/16190/68, A/Beijing/353/89, and A/Netherlands/816/91) strains efficiently agglutinated unmodified chicken and turkey erythrocytes. Importantly, A/H3N2 viruses that emerged after 2010 only agglutinated the 2,6-Sia Poly-LN cells having extended sialylated epitopes. Although some contemporary A/H3N2 viruses can agglutinate turkey erythrocytes when applied undiluted, especially the two 2019 3C3a viruses, extended sialylated LacNAc moieties increased the efficiency of agglutination by 3–10-fold, including the A/H1N1 and influenza B controls. We performed a time course to determine the stability of the glyco-engineered cells, and no loss in titer or autoagglutination for up to three weeks was observed similar to unmodified cells (Supplementary Fig. 6).

The 2,6-Sia Poly-LN cells were employed to antigenically characterize typical recent seasonal A/H3N2 viruses of various clades by HI assay using post-infection ferret sera (Table 2). All antisera showed robust inhibition of the homologous viruses and variable inhibition of heterologous viruses. Egg-derived vaccine strains displayed poor correspondence with data generated with cell-passaged viruses of the same clade. Antisera raised against cell-passaged virus isolates showed greater clade specificity compared to antisera raised against egg-derived vaccine strains. Additionally, egg-derived vaccine strains were inhibited stronger by various antisera than cell-passaged viruses. The HI assay also revealed that antisera raised against recent vaccine viruses, including A/Kansas/14/17 that was selected for the 2019/2020 northern hemisphere influenza vaccine[38], exhibited only minimal cross reactivity against circulating viruses from the same clades, indicating that the circulating viruses differ antigenically from the vaccine strains of the same clade.

The results of the HI assay were compared with a focus reduction assay (FRA) using the same sera and viruses (Supplementary Table 1)[39]. In this assay, the ability of antibodies to block virus infection in mammalian cell culture (MDCK-Siat cells, which overexpress Sia(α2,6)Gal moieties) is quantified. The FRA confirmed the trends observed in the HI assay (Supplementary Fig. 7), indicating that the modified erythrocytes are reliable for antigenic characterization of A/H3N2 viruses.

**Molecular dynamics simulations of HAs in complex with their receptors.** α2,6-Sialyl-Gal is widely considered as the prototypic human receptor for IAVs[17]. All viruses we examined, including A/H1N1 viruses (Supplementary Fig. 8), recognized with high avidity *N*-glycans having an α2,6-linked sialoside on an extended LacNAc moiety. Human respiratory tissue abundantly expresses such extended structures[25,26], and thus we reasoned that mutational changes in recent A/H3N2 viruses led to a reduced binding avidity of the prototypic human receptor (α2,6-sialyl-Gal), which was compensated by making interactions with an extended LacNAc chain.

X-ray crystal structures[31,40,41] have shown that sialic acid is recognized in a conserved hydrophobic pocket (Y98, H183, Y195, and W153) (Supplementary Table 2). It makes further interactions through a hydrogen bonding network with residues 135-137, E190, and S228. Sequence alignments showed that post-2000 strains acquired single point mutations at one of these residues (Supplementary Table 2), which disrupted the hydrogen bonding network, and likely resulted in a reduced binding affinity of the prototypic human receptor, which is a sialoside α2,6-linked to galactose[31]. Furthermore, HAs of early human A/H3N2 strains have Glu at residue 190, which can form a hydrogen bond with O9 of sialic acid, whereas post-2000 strains, favor Asp190 at this position, which due to its shorter side chain, cannot form such an interaction. The 190E/D mutation was accompanied by a 225G/D mutation, which resulted in a rotation of Gal-2 allowing a hydrogen bond interaction with the site chain of 225D[31,40]. This rotation places the extended LacNAc chain closer to the 190-helix and potentially allows for addition interactions. To examine this mode of binding, we compared molecular dynamics generated structures of α2,6-sialyl poly-LacNAc in complex with NL91, NL03, and NL17 (Fig. 3). It recapitulated observations made by X-ray crystallography studies and provided insight in how mutational changes allowed for interactions with an extended LacNAc chains. The MD trajectory of NL91 only showed interactions with sialic acid, including the important hydrogen bond between Sia-1 O9 and Glu190 (Fig. 3a)[31,40]. In the case of NL03 and NL17, Asp190 is at a distance of 4.5 Å to Sia-1 O9, and thus cannot establish a hydrogen bond (Fig. 3d, g). The G225D substitution resulted in a rotation of the bond between the sialic acid and Gal-2 to form an H-bond with its O3[16], (Fig. 3e, h). As anticipated, the resulting rotation of Gal-2 placed the extended LacNAc moieties near the 190-helix[41] resulting in a hydrogen bond between Asp190 and Gal-4 O2, while either Ser or Asn193 provided a H-bond with the acetamide moiety of GlcNAc-3 (Fig. 3e). In addition, Gal-6 (Fig. 3e, h) makes a CH-π interaction

**Table 1 Hemagglutination assay of A/H3N2 virus isolates using unmodified and 2,6-Sia Poly-LacNAc fowl erythrocytes and unmodified guinea pig erythrocytes.**

| Virus | Type | Clade | Unmodified | 2,6-Sia | 2,6-Sia Poly-LN | Unmodified | 2,6-Sia | 2,6-Sia Poly-LN | Unmodified |
|---|---|---|---|---|---|---|---|---|---|
| | | | Chicken | | | Turkey | | | Guinea pig |
| | | | Hemagglutination titers (HAU per 25 µl) | | | | | | |
| A/Bilthoven/16190/68 | H3N2 | Unknown | 24 | 32 | 64 | 64 | 48 | 64 | 48 |
| A/Beijing/353/89 | H3N2 | Unknown | 24 | 64 | 64 | 64 | 64 | 96 | 64 |
| A/Netherlands/816/91 | H3N2 | Unknown | 64 | 128 | 128 | 64 | 128 | 128 | n.t. |
| A/Netherlands/109/03 | H3N2 | Unknown | Neg. | 256 | 256 | 64 | 128 | 128 | n.t. |
| X-161B (A/Wisconsin/067/05) | H3N2 | Unknown | 12 | Neg. | 64 | 32 | 32 | 64 | n.t. |
| IVR-147 (A/Brisbane/10/07) | H3N2 | Unknown | 4 | Neg. | 8 | 16 | Neg. | 8 | n.t. |
| A/Netherlands/761/09 | H3N2 | Unknown | Neg. | Neg. | 64 | Neg. | 64 | 64 | n.t. |
| X-223A (A/Texas/50/12) | H3N2 | 3C.1 | 8 | Neg. | 8 | 12 | Neg. | 8 | n.t. |
| NIB-104 (A/Singapore/INFH-16-0019/16) | H3N2 | 3C.2a1 | 8 | 2 | 16 | 24 | 12 | 24 | 32 |
| A/Netherlands/2413/16 | H3N2 | 3C.2a1 | Neg. | Neg. | 96 | 3 | 24 | 128 | n.t. |
| A/Netherlands/751/17 | H3N2 | 3C.2a1 | Neg. | Neg. | 6 | Neg. | Neg. | 512 | 1 |
| A/Netherlands/757/17 | H3N2 | 3C.2a1 | Neg. | Neg. | 64 | 6 | 16 | 128 | n.t. |
| A/Netherlands/1797/17 | H3N2 | 3C.2a1 | Neg. | Neg. | 32 | Neg. | Neg. | 32 | Neg. |
| A/Netherlands/295/19 | H3N2 | 3C.2a1b | Neg. | Neg. | 192 | Neg. | Neg. | 192 | n.t. |
| A/Netherlands/314/19 | H3N2 | 3C.2a1b | Neg. | Neg. | 128 | Neg. | Neg. | 192 | Neg. |
| A/Netherlands/371/19 | H3N2 | 3C.2a1b | Neg. | Neg. | 32 | Neg. | Neg. | 32 | Neg. |
| A/Netherlands/03466/17 | H3N2 | 3C.2a | Neg. | Neg. | 2048 | Neg. | 6 | 512 | Neg. |
| NIB-112 (A/Switzerland/8060/17) | H3N2 | 3C.2a2 | n.t. | n.t. | n.t. | Neg. | Neg. | 4 | Neg. |
| A/Netherlands/10616/18 | H3N2 | 3C.2a2 | Neg. | Neg. | 3 | Neg. | Neg. | 4 | Neg. |
| A/Netherlands/622/12 | H3N2 | 3C.3 | Neg. | Neg. | 12 | Neg. | Neg. | 12 | 1 |
| A/Switzerland/9715293/13 | H3N2 | 3C.3a | Neg. | Neg. | 16 | 2 | 3 | 24 | 2 |
| A/Netherlands/153/16 | H3N2 | 3C.3a | Neg. | Neg. | 96 | 12 | 24 | 128 | n.t. |
| X-327 (A/Kansas/014/17) | H3N2 | 3C.3a | Neg. | 2 | 16 | 6 | 4 | 16 | 8 |
| A/Netherlands/384/19 | H3N2 | 3C.3a | Neg. | Neg. | 2048 | 12 | 192 | 1536 | 6 |
| A/Netherlands/10002/19 | H3N2 | 3C.3a | Neg. | Neg. | 64 | 12 | 16 | 96 | n.t. |
| A/Netherlands/10006/19 | H3N2 | 3C.3a | Neg. | 2 | 32 | 8 | 12 | 64 | 12 |
| IVR-180 (A/Singapore/GP1908/15) | H1N1 | 6B | 32 | 16 | 48 | 24 | 32 | 24 | 32 |
| BX-69A (B/Maryland/15/15) | B | n.a. | 32 | 16 | 64 | 32 | 32 | 32 | 32 |

*Unknown* clade designation was not used at the time of identification, *n.t.* not tested, *n.a.* not applicable, *neg.* no titer obtained.

**Table 2 Hemagglutination inhibition assay.**

| Virus | Passage | Clade | 3C.2a1a NIB 104 | 3C.2a1b A/NL/314/19 | 3C.2a2 NIB-112 | 3C.2a2 A/NL/3466/17 | 3C.2a2 A/NL/1802/18 | 3C.3a X-327 | 3C.3a A/NL/384/19 |
|---|---|---|---|---|---|---|---|---|---|
| | | | Post-infection ferret sera raised against | | | | | | |
| A/Singapore/INFH-16-0019/16 (NIB 104) | E7Mdck_Siat2hCK | 3C.2a1a | *640* | 320 | 960 | 160 | 480 | <5 | 60 |
| A/Netherlands/1797/17 | Mdck1Siat2hCK | 3C.2a1 | 15 | 160 | 30 | 40 | 80 | <5 | 30 |
| A/Netherlands/314/19 | MdckSiat2hCK | 3C.2a1b | 30 | *240* | 30 | 60 | 80 | <5 | 40 |
| A/Netherlands/10009/19 | mixhCK2 | 3C.2a1b | 20 | 160 | 40 | 120 | 120 | <5 | 30 |
| A/Netherlands/371/19 | MdckSiatmixhCK | 3C.2a1b | <5 | 160 | 5 | 30 | 40 | <5 | 15 |
| A/Switzerland/8060/17 (NIB-112) | E7E1hCK | 3C.2a2 | 640 | 120 | *1920* | 1280 | 240 | <5 | <5 |
| A/Netherlands/3466/17 | Siat1hCK1 10-2 | 3C.2a2 | 15 | 80 | 320 | *480* | 80 | <5 | 30 |
| A/Netherlands/1802/18 | Siat1hCK1 10-1 | 3C.2a2 | 15 | 80 | 40 | 40 | *640* | <5 | 20 |
| A/Netherlands/10616/19 | MdckSiatmixhCK | 3C.2a2 | 15 | 80 | 320 | 640 | 80 | <5 | 20 |
| A/Kansas/14/17 (X-327) | E17E1Siat1hCK | 3C.3a | 640 | <5 | <5 | <5 | <5 | *320* | 60 |
| A/Netherlands/384/19 | Siat2hCK 10-2 | 3C.3a | 30 | 30 | 60 | 40 | 80 | 40 | *240* |
| A/Netherlands/10006/19 | MdckSiatmixhCK2 | 3C.3a | <5 | <5 | 20 | 20 | 30 | <5 | 240 |

Italic values indicate homologous titers/antiserum virus.

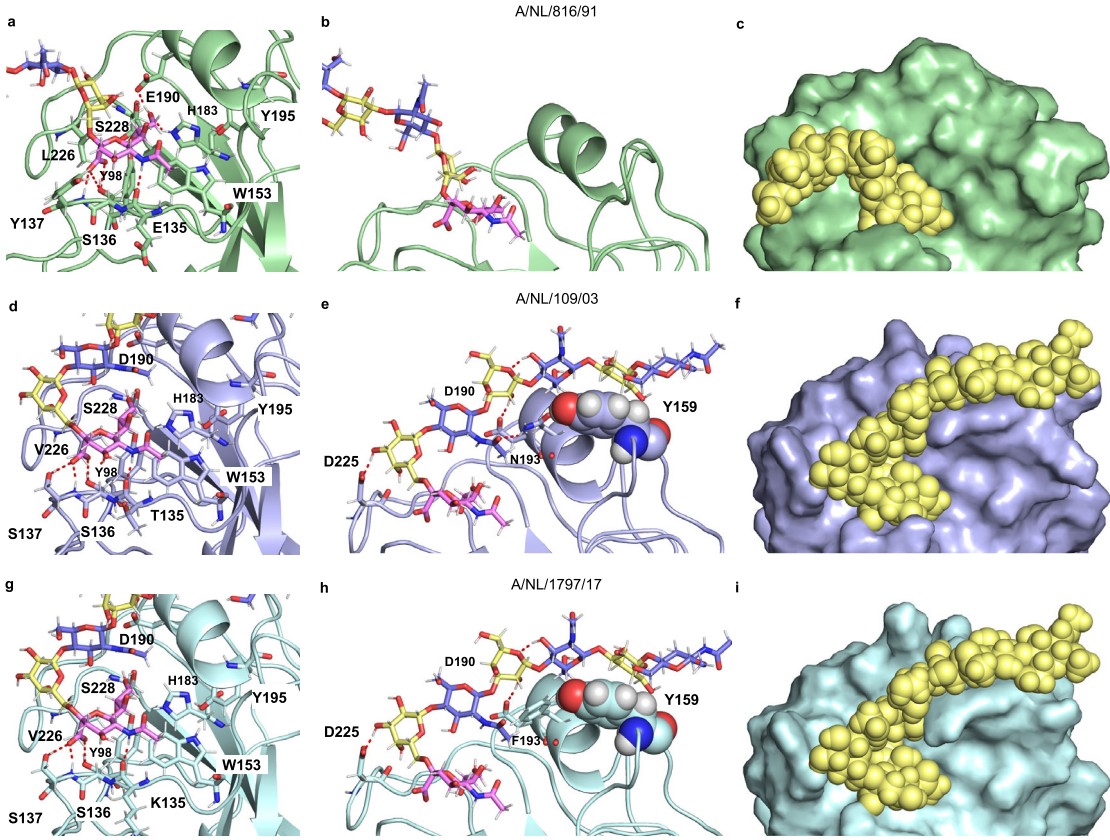

**Fig. 3 Structural comparison of HAs from evolutionary different strains in complex with extended glycan receptor.** Details of the sialic acid binding sites are shown for A/NL/816/91 (**a**), A/NL/109/03 (**d**), and A/NL/1797/17 (**g**). The interactions of the poly-LN chain with the protein are shown for A/NL/816/91 (**b**), A/NL/109/03 (**e**), and A/NL/1797/17 (**h**). The surface and spheres representations of HA/glycans complexes are shown for A/NL/816/91 (**c**), A/NL/109/03 (**f**), and A/NL/1797/17 (**i**).

with the side chain of Y159 (Supplementary Fig. 10). Such interactions are often observed in glycan-protein complexes and contribute substantially to binding[42]. Structural analysis of the HAs of NL91 and NL03 showed that mutations distal to the receptor-binding domain (A131T, H155T, and E156H), reorient the side chain of Y159 resulting in an extended receptor-binding site allowing interactions with Gal-6 (Supplementary Fig. 9). The MD simulations support that A/H3N2 of the 3C.2 clade have undergone mutations to create and extended binding site to compensate for reduced binding of the non-extended human receptor (Fig. 3c, f, i). Interestingly, 3C.3a viruses, which can utilize shorter receptors having two LacNAc units (Fig. 1a), have a tyrosine to serine mutation at position 159 (Supplementary Table 2). Thus, the dependence on extended receptors is reversible, and these viruses have found a way to bind shorter structures with sufficient affinity for infection. The observation that distant mutations can alter the position of a side chain of an amino acid in the receptor-binding site, which subsequently can become susceptible to antigenic pressure, indicates that such remote mutations need to be considered for the evolution of receptor binding and antigenic distance.

## Discussion

Changes in receptor-binding properties of H3N2 viruses were first noticed by a lack of agglutination of commonly employed red blood cells and poor recovery of isolated viruses that were propagated in laboratory hosts such as eggs and MDCK cells[29,43]. Conventional binding assays using a limited set of relevant glycans showed these viruses had lost the ability to bind to what was regarded as the canonical human-type receptors, namely α2,6-sialyl-Gal[21,44].

Screening of a glycan microarray populated with symmetrical *N*-glycans (each arm modified by the same epitope) identified receptors for contemporary A/H3N2. In particular, these viruses could bind bi-antennary *N*-glycans having an extended sialo-LaNAc epitope at each of its arm. Modeling studies indicated that the two sialic acid moieties are sufficiently spaced to bind to a protomer of the same HA trimer. It was proposed that the resulting bidentate binding mode increased the binding avidity[22].

Progress in methods for the chemoenzymatic synthesis of glycans[45] allowed us to construct a glycan microarray populated with bi-antennary *N*-glycans that more closely resemble structures expressed in human respiratory tissue[25,26], and include asymmetrical glycans having either one or two sialic acid moieties linked to LacNAc chains of different length. It made it possible, for the first time, to uncover the minimal receptor of contemporary A/H3N2 viruses, which is a bi-antennary *N*-glycan in which one of the arms is extended by three consecutive LacNAc units that is modified by an α2,6-sialoside. It demonstrated that the previously proposed bidentate binding event does not substantially contribute to binding[22]. Based on our microarray data, sequence alignments, reported crystal structures, and MD simulations, we developed alternative binding model for A/H3N2 of the 3C.2 clade. In this model, mutations remote to the receptor-binding domain resulted in the rotation of the side chain of tyrosine at position 159 which gives rise to an extended binding site. Furthermore, a 225G/D mutation reoriented the poly-LacNAc chain allowing it to make interactions with the extended binding site. These interactions compensate for reduced contacts with sialic acid caused by mutations in the receptor-binding site arisen from antigenic pressure[4]. It is, however, possible that

factors other than antigenic pressure may have contributed to adaptions of receptor specificity. For example, a binding requirement for sialic acid on poly-LacNAc chain may provide a benefit because such structures are usually not found on *O*-glycans of mucins, and therefore such a receptor specificity may contribute to escaping the mucosal barrier".

The identification of the minimal receptor for contemporary H3N2 viruses made it possible to engineer the surface of erythrocytes with functional receptors allowing easy antigenic characterization of recent A/H3N2 isolates using the HI assay. It confirmed that antigenically distinct viruses are circulating in humans and that egg-passaged A/H3N2 vaccine components match poorly to circulating strains. Due to the failure of the classical HI, focus reduction assays have been employed to antigenically characterize A/H3N2. These assays are, however, time-consuming, have low throughput, suffer from low reproducibility, and can lead to adapted mutations thereby providing incorrect results[12]. These limitations are addressed by the glyco-engineered red blood cells described here. The altered receptor requirement of A/H3N2 viruses is also complicating propagation in the laboratory. Recently, an MDCK cell line (hCK) was introduced in which α2,3-sialyl transferases were genetically removed and ST6Gal1, which introduces human-type receptors, was overexpressed[33]. This cell line supports efficient replication of contemporary human A/H3N2 viruses that maintain higher genetic stability[46]. The minimal receptor requirements of contemporary A/H3N2 viruses established in this study support further engineering of such cells to biosynthesize extended α2,6-sialylated LacNAc moieties. Furthermore, it is the expectation that the described glycan engineering approach for erythrocytes can be extended to other cells, such as MDCK cells, to quickly provide laboratory hosts for virus replication. The results presented here may also provide a rationale for why certain sialylated inhibitors do not bind HA when expressed in CHO *vs*. HEK cells where the nature of the sialylated linkages are different[47]. Finally, an understanding of the evolution of receptor specificities of A/H3N2 viruses and associated antigenic changes will facilitate the development of predictive evolutionary models for the reliable selection of vaccine strains.

## Methods

### Virus production
*Materials*. Eagle's minimal essential medium (EMEM), penicillin, streptomycin, L-glutamine, sodium bicarbonate, HEPES, 1x non-essential amino acids, and *N*-tosyl-L-phenylalanine chloromethyl ketone (TPCK) treated trypsin were purchased at Lonza Benelux BV, Breda, The Netherlands. Fetal bovine serum was obtained from Greiner.

Madin-Darby canine kidney (MDCK), MDCK-Siat, and hCK (a humanized MDCK knock-out cell line, deficient of several 2,3-specific sialyltransferase enzymes, and overexpressing human-type 2,6-sialylated receptors)[33]. cells were cultured in Eagle's minimal essential medium supplemented with 10% fetal bovine serum (FBS), 100 U mL$^{-1}$ penicillin (P), 100 U mL$^{-1}$ streptomycin (S), 2 mM L-glutamine (L-glu), 1.5 mg mL$^{-1}$ sodium bicarbonate (NaHCO$_3$), 10 mM HEPES, and 1x non-essential amino acids (NEAA)[33]. In addition, hCK cells were supplemented with 2 µg mL$^{-1}$ puromycin and 10 µg mL$^{-1}$ blasticidin and MDCK-Siat cells were supplemented with 1 mg mL$^{-1}$ Geneticine. To produce virus stocks, cells were washed twice with PBS 1 h after inoculation with the virus of interest and cultured in infection media, consisting of EMEM supplemented with 100 U mL$^{-1}$ penicillin, 100 µg mL$^{-1}$ streptomycin, 2 mM glutamine, 1.5 mg mL$^{-1}$ sodium bicarbonate, 10 mM Hepes, 1x non-essential amino acids, and 20 µg mL$^{-1}$ N-tosyl-L-phenylalanine chloromethyl ketone (TPCK) treated trypsin. To produce virus stocks in eggs, 100 µL of virus was inoculated in the allantoic cavities of 11-day-old embryonated hens' eggs. The allantoic fluid was harvested after 2 days.

### Microarray studies
*Materials*. Virus isolates were produced as described above. Oseltamivir was purchased from Sigma Aldrich [Cat# SML1606]. CR8020 A/H3N2 stem antibody was kindly provided by Dr. Dirk Eggink and expressed following previously published procedures[48]. Goat anti-human Alexa-647 [Cat# A21445] and streptavidin-AlexaFluor 635 [Cat# SA1011] antibodies were obtained from Thermo Fisher. Control lectins *Erythrina cristagalli* agglutinin (ECA) [Cat#B-1145], *Sambuca nigra*

agglutinin (SNA) [Cat# B-1305], and *Maackia amurensis* lectin I (Mal-I) [Cat# B-1315] were purchased from Vector Labs.

*Arrayer and printing surfaces*. Compounds were printed on amine reactive, NHS-activated glass slides (NEXTERION ® Slide H) from Schott Inc exploiting the free amine of an asparagine moiety at the reducing end of the *N*-glycans. Scienion sciFLEXARRAYER S3 non-contact microarray printer equipped with a Scienion PDC80 nozzle (Scienion Inc) was used for printing. Glycans were dissolved in printing buffer (sodium phosphate, 250 mM, pH 8.5) at a concentration of 100 µM. Each compound was printed in replicates of 6 with a spot volume of ~400 pL, at 20 °C and 50% humidity. Slides were blocked with 5 mM ethanolamine in Tris buffer (pH 9, 50 mM) for 1 h at 50 °C and rinsed with DI water after printing.

*Glycan microarray*. Quality control was performed using the plant lectins ECA (specific for terminal Gal), SNA (specific for α2,6-linked Neu5Ac), and MAL-I (specific for α2,3-linked Neu5Ac) and is shown in Supplementary Fig. 1. Quality control of the CR8020 A/H3N2 influenza hemagglutinin stem specific antibody specificity was performed by incubation of the antibody to the array as described below, in the absence of a virus (Supplementary Fig. 1). The printed library of compounds comprised the glycans described in the Supplementary Information (#8, #9, #11, #12, #13, #15, #16, #17) and published previously (#1-#7, #10, #14)[49].

*Sample*. Virus isolates (25 µL) were diluted with PBS-T (PBS + 0.1% Tween, 25 µL) and applied to the array surface in the presence of oseltamivir (200 nM) in a humidified chamber for 1 h, followed by successive rinsing with PBS-T (PBS + 0.1% Tween), PBS, and deionized water (2x) and dried by centrifugation. The virus-bound slide was incubated for 1 h with the CR8020 A/H3N2 influenza hemaglutinin stem specific antibody (100 µL, 5 µg mL$^{-1}$ in PBS-T) and washed according to previous washing procedure. A secondary goat anti-human Alexa-Fluor-647 antibody (100 µL, 2 µg mL$^{-1}$ in PBS-T) was applied, incubated for 1 h in a humidified chamber and washed again as described above. The control lectins containing a biotin tag were visualized with Streptavidin-AlexaFluor635. Slides were dried by centrifugation after the washing step and scanned immediately.

*Detection and data processing*. The slides were scanned using an Innopsys Innoscan 710 microarray scanner at the appropriate excitation wavelength. To ensure that all signals were in the linear range of the scanner's detector and to avoid any saturation of the signals various gains and PMT values were employed. Images were analyzed with Mapix software (version 8.1.0 Innopsys) and processed with our home written Excel macro. The average fluorescence intensity and SD were measured for each compound after exclusion of the highest and lowest intensities from the spot replicates (n = 4).

### Glycomic analysis
*Materials*. Acetic acid [Cat# 5330010050], MS–grade formic acid [Cat# 5330020050], 2-aminoanthrallic acid (2-AA) [Cat#10680], dimethyl sulfoxide (DMSO) [Cat# W387520], and sodium cyanoborohydride [Cat# 156159] were obtained from Sigma Aldrich. Trifluoro acetic acid (TFA) was purchased from Acros Organics [Cat#434161000]. MS-grade acetonitrile (MeCN) was obtained from Biosolve [Cat# 200-835-2]. PNGase F was purchased from Roche Diagnostics (1U defined as the amount of enzyme catalysing the conversion of 1 µmol(substrate) min$^{-1}$) [Cat#06538355103]. Denaturation buffer (0.5% SDS, 40 mM dithiothreitol (DTT)), 1% NP-40, and glycobuffer (50 mM sodium phosphate) were obtained from New England BioLabs. Ammonium formate was purchased from Fluka chemicals [Cat# AGG1946-85021], C18 solid phase extraction (SPE) Sep-Pak® Vac (1cc) columns from Waters Corporation [Cat# WAT054955], PGC SPE Hypercarb Hypersep (1cc) columns from Thermo Scientific [Cat# 60106-303], and PD Minitrap Sephadex G-10 size exclusion cartridges from GE Healthcare [Cat# 28-9180-10]. MilliQ water was obtained from a Synergy® water purification system.

*N-glycan extraction and release from erythrocytes*. Cell surface *N*-glycans were extracted according to a reported protocol[35]. Briefly, erythrocytes (400 µL, 50%) were concentrated by centrifugation (430 rcf, 10 min) and removal of the supernatant. Erythrocytes were lysed under gentle shaking at room temperature for 1 h using deionized water (3x pellet size). The suspension was centrifuged (4000 rcf, 10 min), the supernatant removed, and the pellet resuspended in deionized water. The process was repeated until the pellet decolorized, indicating an efficient lysis of the erythrocytes. Denaturing of the cell membrane pellet was performed by heating for 10 min to 95 °C in denaturation buffer (0.5% SDS, 40 mM DTT in H$_2$O). The *N*-glycans were released during overnight incubation at 37 °C with PNGase F (5 U) in a sodium phosphate buffer (50 mM, pH 7.5) or with Endo F2 in sodium acetate buffer (50 mM, pH 4), both containing NP-40 (1%).

*Purification and labeling of N-glycans*. The released *N*-glycans were applied to a C18 SPE cartridge and glycans were eluted with 5% MeCN in H$_2$O (0.05% TFA, 1 mL). The eluate was further purified on a PGC SPE cartridge by gradually increasing the hydrophobicity from 100% H$_2$O (0.05% TFA, 1 mL) to 5% MeCN in H$_2$O (0.05% TFA, 1 mL), and to 50% MeCN in H$_2$O (0.05% TFA, 1 mL) which eluted the glycans. After drying in a N$_2$ flow, the sample was dissolved in H$_2$O

(10 μL) and labeled by addition of a solution (10 μL) of 2-AA (48 mg mL$^{-1}$) and sodium cyanoborohydride (63 mg mL$^{-1}$) in DMSO/acetic acid (10:3 v/v) for 2 h at 65 °C. The crude mixture was diluted with H$_2$O (80 μL) and purified on a Minitrap Sephadex G-10 gravity column by washing with of H$_2$O (700 μL) and eluting with H$_2$O (600 μL). The eluate was dried under N$_2$ flow and dissolved in 20 μL of 50% MeCN in H$_2$O prior to LC-MS analysis.

*HILIC-IMS-QTOF analysis of N-glycans and data treatment.* The *N*-glycan analysis was performed on a 1260 Infinity liquid chromatography system (Agilent Technologies) coupled to a 6560 IM-QTOF mass spectrometer (Agilent Technologies) and HPLC separation with a ZIC®-cHILIC column (3 μm, 100 Å, 150 mm × 2.1 mm, Merck) and a similar guard column (20 mm × 2.1 mm, Merck) using a gradient from 30% A to 50% A within 25 min (A: 50 mM (NH$_4$)$_2$CO$_3$ in H$_2$O; B: MeCN; 0.2 mL min$^{-1}$; 60 °C). MS analysis was performed with a drying gas temperature of 300 °C and a flow of 8 mL min$^{-1}$. The nebulizer pressure was set to 40 psi, the sheath gas flow to 11 L min$^{-1}$ and the temperature to 350 °C. Measurements were run in negative mode with the capillary established at 3500 V. During the runs the Agilent tuning mix was infused for mass calibration based on reference signals at m/z 112.9855 and *m/z* 1033.9881.

The data analysis was performed with the Mass Hunter IM-MS Browser and the find feature function filtering for masses with an ion intensity of ≥500. Found masses were processed with the online Glycomod tool to identify glycan related masses[50].

### Glycoengineering of erythrocytes
*Materials.* *Arthrobacter ureafaciens* neuraminidase was purchased at New England Biolabs (1U defined as the amount of enzyme catalyzing the conversion of 1 μmol(substrate) min$^{-1}$) [Cat# P0722L]. Mammalian glycosyltransferases were expressed according to literature reports[37]. B3GnT2 and ST6Gal1 were cleaved and purified from GFP tags prior to use. Alkaline phosphatase (FastAP) was purchased at Thermo Scientific [Cat# EF0651]. Nucleotide sugars UDP-Gal, UDP-GlcNAc and CMP-Neu5Ac were obtained from Roche Diagnostics [UDP-Gal: Cat# 07703562103; UDP-GlcNAc: Cat# 06369855103; CMP-NeuAc: Cat# 05974003103].

*Erythrocyte preparation.* Fresh blood from chicken or turkey was centrifuged (10 min, 430 rcf) followed by removal of the supernatant. Pellets were washed three times in PBS with intermittent centrifugation (430 rcf, 10 min). Erythrocyte solutions were stored in a 50% solution in PBS until further use.

*Enzymatic extension.* To a suspension of fowl erythrocytes (250 μL, 50%), PBS (900 μL) and *Arthrobacter ureafaciens* neuraminidase (12 U) were added. The cells were incubated for 6 h at 37 °C while tilting. Next, glycosyltransferases B4GALT1 (37.5 μL, 1 mg mL$^{-1}$) and B3GnT2 (37,5 μL, 1 mg mL$^{-1}$), the nucleotide sugars UDP-Gal (4.4 mM) and UDP-GlcNAc (4.4 mM), alkaline phosphatase (6 U), MnCl$_2$ (2 mM) and BSA (6 μL, 2 mg mL$^{-1}$) were added. This reaction mixture was incubated overnight at 37 °C while tilting. The erythrocytes were washed in PBS (2x, 600 μL) and the pellet was reconstituted in 900 μL PBS. Resialylation of the erythrocytes was performed using ST6Gal1 (37.5 μL, 1 mg mL$^{-1}$) and CMP-Neu5Ac (4.4 mM) in the presence of alkaline phosphatase (6 U) and BSA (6 μL, 2 mg mL$^{-1}$) for 4 h at 37 °C while tilting. The erythrocytes were washed in PBS (1x, 600 μL) and diluted to a 1% solution for hemagglutination (inhibition) assays.

**Stability assay**. Fresh fowl erythrocytes were glyco-engineered as described above or used unmodified. A hemagglutination assay was performed every 2 days using two viruses, A/NL/761/09 and A/NL/816/91. Hemagglutination assays were performed as described below in full biological triplicates. The means were plotted ±SEM and are shown in Supplementary Fig. 6.

**Hemagglutination assay**. Hemagglutination assays were performed following standard procedures[7]. Briefly, virus stocks were two-fold serial diluted in the presence of oseltamivir (20 nM). Turkey erythrocytes (1%, 25 μL) were mixed with the serial diluted viruses and incubated for 1 h at 4 °C before recording of the results. Titers were expressed as the highest dilution of virus stock that completely agglutinated the turkey erythrocytes.

**Hemagglutination inhibition assay**. Hemagglutination inhibition assays were performed following standard protocols[7]. Briefly, for the preparation of the antisera, ferrets were inoculated intranasally and blood was obtained 14 days later. Experiments were performed in strict compliance with European guidelines (EU Directive on Animal Testing 86/609/EEC) and Dutch legislation (Experiments on Animals Act, 1997). An independent animal experimentation ethical review committee (Dutch Stichting Dier Experimenten Commissie Consult) approved all animal studies (license AVD101002015340). Antisera were pre-treated with receptor destroying enzyme (RDE) by incubating overnight with an in-house produced filtrate of *Vibrio cholera* at 37 °C followed by 1 h incubation at 56 °C. The treated antisera were pre-absorbed with extended turkey erythrocytes (10%) in two cycles of 1 h incubation at 4 °C. Pre-absorbed antisera were two-fold serial diluted

(starting at 1:10) and mixed with virus stock (25 μL) containing 4 hemagglutinating units. Viruses were incubated with the antisera for 1 h at 4 °C in the presence of BSA (0.25%) and oseltamivir (20 nM). Turkey erythrocyte solution (25 μL, 1%) was added and after 1 h incubation at 4 °C inhibition patterns were recorded. Titers were expressed as the value of the highest serum dilution that gave complete inhibition of agglutination.

**Focus reduction assay**. Focus reduction assays were performed following standard protocols[51]. First, infectious titers of the virus stocks were determined in hCK cells as described previously[52]. RDE-treated sera were twofold diluted in a 96-well plate (starting at 1:10) and mixed 1:1 with 100 TCID$_{50}$/50 μL of virus. After 1 h incubation at 35 °C, 100 μL of the mixtures were transferred to hCK cells and after 90 min incubation at 35 °C, cells were washed and overlaid with 1.6% carboxymethylcellulose. After 48 h at 35 °C, cells were washed and fixed with formalin and permeabilized using 0.5% Triton X-100 for 10 min at room temperature. Subsequently, immunostaining was performed using a mouse monoclonal antibody (HB65; EVL, Woerden, The Netherlands) directed against the viral nucleoprotein (NP), followed by a horseradish peroxidase-labeled goat anti mouse immunoglobulin preparation (GAM-HRPO, Invitrogen, Foster city, CA), both for 1 h at room temperature. After washing, True-Blue substrate (KPL, Gaitherburg, Maryland) was added followed by a 10 min incubation at room temperature. The plates were washed, dried, and submitted to automated image capture using a Series 6 ImmunoSpot Image Analyzer (CTL Immuno-Spot, Cleveland OH, USA) to quantitate the percentage well area covered by spots of infected cells. Inhibition ≥90% was considered positive for neutralization.

### Structural studies
*Alignment.* To investigate the relevance of specific mutations in defining receptor preferences during antigenic drift, primary amino acid sequences were compared of A/H3N2 HAs from 1968 to 2019. Protein sequences and structures were derived from the 3DPlu database (http://3dflu.cent.uw.edu.pl/index.html)[53] and the GISAID webpage (https://www.gisaid.org). It identified specific mutations that may influence receptor-binding preferences. We paid specific attention to mutations in the four structural elements that define the RBS, including the 130-loop, the 150-loop, the 190-helix, and the 220-loop. The results from this analysis are summarized in Supplementary Table 2.

*MD simulations.* A starting pose of the NL91 from 1991 was generated by superimposition of the model derived structure (id code ACU12494) [http://3dflu.cent.uw.edu.pl/index.html] onto the X-ray crystal structure of the A/HK/1/1968 H3N2 influenza virus hemagglutinin in complex with 6′-SLNLN (pdb code 6TZB)[41]. The starting pose of NL03 was generated by using the X-ray crystal structure of the A/Wy/3/03 influenza virus hemagglutinin in complex with 6′-SLN (pdb code 6BKR)[37]. Glycan receptors, NeuAcα2-6(LacNAc)$_2$ and NeuAcα2-6(LacNAc)$_3$, were generated by using the carbohydrate builder GLYCAM-web site [http://glycam.org]. The glycosidic torsion angles of the monosaccharides were maintained as observed by X-ray crystallography, while those not resolved were defined according to the lower energy values predicted by the GLYCAM-web modeling tool. The structural model of the NL17 was obtained by using the mutagenesis tool implemented in PyMOL. The resulting poses were used as starting points for molecular dynamics (MD) simulations. The MD simulations were performed using the Amber16 program4 with the protein.ff14SB, the GLYCAM_06j-1 and the water.tip3p force fields parameters. Next, the starting 3D geometries were placed into a 10 Å octahedral box of explicit TIP3P waters, and counter ions were added to maintain electroneutrality. Two consecutive minimization steps were performed involving (1) only the water molecules and ions and (2) the whole system with a higher number of cycles, using the steepest descent algorithm. The system was subjected to two rapid molecular dynamic simulations (heating and equilibration). The equilibrated structures were the starting points for a final MD simulations at constant temperature (300 K) and pressure (1 atm). In all, 100 ns Molecular dynamics simulations without constraints were recorded, using an NPT ensemble with periodic boundary conditions, a cut-off of 10 Å, and the particle mesh Ewald method. A total of 50,000,000 molecular dynamics steps were run with a time step of 1 fs per step. Coordinates and energy values were recorded every 50000 steps (50 ps) for a total of 1000 MD models. The detailed analysis of the H-bond and CH-π interactions was performed along the MD trajectory using the cpptraj module included in Amber-Tools 16 package.

**Reporting summary**. Further information on research design is available in the Nature Research Reporting Summary linked to this article.

## Data availability
The data supporting the findings in this paper are available in the manuscript and Supporting Information. A full overview of the identified glycomic structures can be found in Supplementary data 1 and Supplementary data 2. Source data are provided with this paper. Sharing materials described in this work will be subject to standard material transfer agreements. Source data are provided with this paper.

## Code availability

The script for Microsoft Excel Macro for batch processing glycan microarray data is uploaded to https://github.com/enthalpyliu/carbohydrate-microarray-processing. https://doi.org/10.5281/zenodo.5146251

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

## Acknowledgements

This project is supported by the Netherlands Organization for Scientific Research (NWO TOPPUNT 718.015.003) to G.-J.B., R.P.dV. is a recipient of ERC starting grant 802780 and a Beijerinck Premium of the Royal Dutch Academy of Sciences. S.H. is supported by NWO VIDI grant 91715372, R.A.F. by NIAID/NIH contract HHSN272201400008C, and K.W.M and G.-J.B. by U.S. National Institutes of Health grants (R01GM130915, P41GM103390, and U01GM120408). We thank T. Manders, dr. H.G.R. Matthijs and dr. R.M. Dwars (Faculty of Veterinary Sciences, Utrecht University) for providing chicken blood and M. Pronk and R. van Beek (Department of Viroscience, Erasmus MC) for technical assistance. Dr. L. Liu (CCRC) and dr. M.A. Wolfert (Utrecht University) developed, printed, and validated the glycan microarray. We would like to thank Dirk Eggink from the Amsterdam Medical Center for supplying the CR8020 antibody.

## Author contributions

G.-J.B., R.P.dV., F.B., and R.J.vB. designed the project; R.J.vB., F.B., L.U., T.B., and R.P.dV. performed experiments and data analysis; R.A.F., G.-J.B., S.H., and R.P.dV. provided scientific guidance on experimental setup and data interpretation; K.W.M., D.C., and J.Y.Y. provided recombinant enzymes; G.-J.B., R.J.vB., F.B., L.U., S.H., R.A.F., and R.P.dV. wrote the manuscript; and all authors provided comments and suggestions on the manuscript.

## Competing interests

F.B., R.J.vB., R.P.dV., and G.-J.B. are inventors for a patent application for the glyco-engineered red blood cells described in this publication for antigenic charterization of *Influenza viruses*. Title: means and methods for detecting, producing, isolating, or characterizing influenza. Application number: EP20179197. Status: provisional application.

## Ethics statement

Animal studies were performed in accordance to relevant guidelines and approved by an independent animal experimentation ethical review committee 'stichting DEC consult'.
