## [Peer Review File · Nature Communications]

Reviewer comments, first round -

Reviewer #1 (Remarks to the Author):

The H3N2 human influenza viruses began to show reduced binding to erythrocytes that are used in standard HA assays about 20 years ago, and the basic mechanisms associated with that process were revealed to be due to a couple of key mutations in the sialic acid binding site of the virus HA. The finding that extended and multi-antennary glycans were the favored receptor was also reported a number of years ago. A basic factor that underlies this study is that the main method for distinguishing the antigenic variation of the influenza viruses is still the HI assay, which was developed around 90 years ago. The lack of HA activity of the recent H3N2 viruses has therefore interfered with the easy use of HI assays to determine, for example, which strains to include in vaccines.

In this study the authors use this background information to develop new methods for preparing erythrocytes for performing HI assays with the modern strains of H3N2 human influenza. The study is over all quite well done and comprehensive, and is a very useful addition to our understanding of the issues associated with the binding of influenza to its receptors. There is not a lot of new biological insight into the biological selection that has occurred to give rise to these HA types, or how the selection occurred through immune or receptor variation, or what was driving this phenomenon.

It was rather difficult to understand the data in detail, in part due to the complexity of the studies, and some of the supplementary files did not seem to open, but overall the studies appear to be well conducted and the end results are clear enough.

There are some things that may be worth addressing.

- 1) Page 1, line 20; Page 3, Line 23 (and elsewhere) - is it completely clear what the selection was that resulted in the change - it is often stated that this was only immune selection, but there may be a benefit for the virus to use the altered receptors? Eggs and MDCK cells are distinct from the human respiratory tract so the connection to the natural tissues seems important. There are no references given to support this statement at the end of page 3.
- 2) Page 2, Line 20/21; Page 3, Line 7+ - some more specific references to the relevance of the HI test for understanding immunity and vaccine strain selection, and why the field did not develop another assay that avoided the problems of the HI - since this is the rationale for this study.
- 3) Page 3, Line 1 - the H3N2 viruses are not generally associated with high mortality...or define what this means.
- 4) Page 5, Line 9+ - why are those erythrocytes being studied - what is the reason for comparing chicken and turkey (are those used in the standard WHO-approved HI?); why are guinea pig RBCs included? - those are not analyzed later on in detail.
- 5) Page 7, Line 4+ - The comparison of the chicken and turkey RBC glycomics data were hard to compare as presented - maybe the key difference(s) that influence influenza binding can be highlighted - In Fig. 1 and elsewhere.
- 6) Page 11, Line 2 and 4 - "human receptor" is not a useful term when it is used to mean alpha2,6-linked Sia, and the more complex extended glycans in the human respiratory tract. More more specific designations need to be given.
- 7) Page 11, Line 7 - what is the prototypic human receptor?
- 8) Page 12, Line 16 - this is Conclusions? Is this necessary if it basically repeats the abstract?
- 9) Page 14, Line 7 - describe the hCK cells.

Reviewer #2 (Remarks to the Author):

This paper describes the change of glycan binding specificity of A/H3N2 influenza viruses from

regular 2,-6-linked sialyl N-glycans to the glycans with extended LacNAc repeats capped with 2,6-linked sialic acid, and as such the characterization of A/H3N2 subtypes and variants become a problem because the new virus variants cannot agglutinate erythrocytes. This important finding is confirmed by elegant synthesis of various N-glycans particularly those with extended LacNAc terminated with sialylation and by the glycan array analysis at the protein and virus levels. It was further confirmed by glycoengineering of erythrocytes to install such extended structures on the cell surface and demonstrated their restored agglutination activity and thus making possible their characterization. Finally, molecular modeling was performed to rationalize the change of binding specificity which led to the identification of specific amino acid change to cause the loss of receptor binding.

Overall, the finding is very important and the work should provide a new understanding of influenza virus evolution and a better characterization of newly evolved viruses so a quick and accurate identification of new variant and vaccine design can be done in time to control the spread of the virus. This paper is certainly acceptable for publication.

One minor point is suggested to improve the clarity of some descriptions before publication. The N-glycans synthesized all have the Asn residue at the reducing end. However, in the description of glycan array preparation, all N-glycans used for printing on the NHS glass surface have GlcNAc at the reducing end. This should be clarified and how the glycan reacted with the active ester on the glass surface should be stated. Another point is that the authors stated that airway epithelial cells contain extended LacNAc repeats on the cell surface. It would be better to clearly describe what type of extended structures and relative affinity for different viral hemagglutinins have been reported in the literature so the readers can have a better appreciation of the contributions from this group.

Reviewer #3 (Remarks to the Author):

The manuscript by Broszeit et al is important for the development of assays needed to develop vaccines for H3N2 type influenza viruses that have lost the ability to agglutinate fowl erythrocytes.

Although the focus is fowl erythrocytes, I would like some discussion on the relevance to human erythrocytes and glycans found on the respiratory track. Human erythrocytes are also used in the HIA assay. Most human glycans contain core fucose and multiple extended GlcNAc's do NOT occur (PLoS Pathog 9(3): e1003223. doi:10.1371/journal.ppat.1003223). Do the authors have human glycan data of relevance for human virus interactions that they could include? The data may also explain why certain sialylated inhibitors do not bind HA when expressed in CHO versus HEK cells where the nature of the sialylated linkages are different (J Immunol. 2020 Feb 15;204(4):1022-1034. doi: 10.4049/jimmunol.1901145. Epub 2020 Jan 6.).

Please expand the discussion to include these studies and the implication for human virus interactions.

The science is excellent and I recommend publication with these changes.

Minor corrections

page 5 line 8, add anti-H3 antibody as this is what is shown in Fig 1A.
page 12, should be titled discussion not introduction.

We thank the referees for their supporting statements and constructive remarks to improve the clarity of the manuscript. As described in detail below, all issues have been addressed in the revised manuscript.

Referee 1.

Remark. Page 1, line 20; Page 3, Line 23 (and elsewhere) – is it completely clear what the selection was that resulted in the change – it is often stated that this was only immune selection, but there may be a benefit for the virus to use the altered receptors? Eggs and MDCK cells are distinct from the human respiratory tract so the connection to the natural tissues seems important. There are no references given to support this statement at the end of page 3.

Response. We thank the reviewer for the insight. It is widely accepted that neutralizing antibodies target the receptor binding site of the HA protein, which has resulted in mutations at this region of the protein to avoid immune detection. A seminal paper supporting this notion is by Koel *et al.* entitled “Substitutions near the receptor binding site determine major antigenic change during influenza virus evolution. *Science* **342**, 976-979 (2013)” (ref. 3 of this paper). It demonstrates that major antigenic changes can be caused by single amino acid substitutions near the receptor binding site. These substitutions substantially skew the way the immune system detects the virus. The final section of our paper, dealing with molecular dynamics simulations of HAs in complex with receptors, shows that these mutations have resulted in fewer interactions with the sialic acid moiety. To compensate for this loss of interaction, further mutations have occurred that created and extended binding site that allows interactions with a poly-LacNAc chain. Previously reported structural studies combined with the sequence alignment data and MD simulations indicate that viruses preceding 2000, only recognize 2,6-linked sialyl-Gal and did not have a requirement for extended LacNAc epitopes. The requirement for sialosides presented on an extended LacNAc moiety occurred at a later stage of evolution.

Our paper does not exclude the possibility that factors other than antigenic pressure may have contributed to changes in receptor specificity. It is written from a perspective of immune pressure, which is the primary driver of alterations near the receptor binding sites. To address the concern of the referee, the conclusion section includes a statement that factors other than immune pressure may have contributed to receptor adaptation: “*It is, however, possible that factors other than antigenic pressure may have contributed to adaptations of receptor specificity. For example, a binding requirement for sialic acid on poly LacNAc moieties may provide a benefit because such structures are usually not found on O-glycans of mucins, and therefore such a receptor specificity may contribute to escaping the mucosal barrier.*”

The referee note that the following statement is not supported by references. “*Thus, failure of contemporary A/H3N2 viruses to agglutinate erythrocytes, the poor replication in mammalian cells and the emergence of egg adaptive mutations are probably due to an adaptation to glycan receptors that are not expressed by these cell substrates.*” This statement should be regarded as the hypothesis that has been tested in our paper. The preceding text, which is ample supported by references, provides a rationale for the hypothesis.

Remark. Page 2, Line 20/21; Page 3, Line 7+ - some more specific references to the relevance of the HI test for understanding immunity and vaccine strain selection, and why the field did not develop another assay that avoided the problems of the HI - since this is the rationale for this study.

Response. Additional references have been added to support the relevance of the HI assay for understanding immunity and vaccine selection (ref 8-12). Furthermore, WHO reports highlight the importance of antigenic surveillance by HI of circulating influenza strains (see ref. 7).

The discussion section describes efforts to develop alternative assays for HI. It states “*Due to the failure of the classical HI, focus reduction assays have been employed to antigenically characterize A/H3N2. These assays are, however, time-consuming, have low throughput, suffer from low*

reproducibility, and can lead to adapted mutations thereby providing incorrect results¹². These limitations are addressed by the glyco-engineered red blood cells described here.”

Remark. Page 3, Line 1 - the H3N2 viruses are not generally associated with high mortality...or define what this means.

Response. The text has been rephrased and a reference has been added. It reads as follows; “*A/H3N2 viruses, which have become the leading cause of seasonal influenza illness and death since their introduction in the human population in 1968^{13,14}, exhibit a particularly rapid antigenic drift. As a result, the WHO has recommended 28 vaccine strain updates since these viruses started circulating in humans¹³.*”

Remark. Page 5, Line 9+ - why are those erythrocytes being studied - what is the reason for comparing chicken and turkey (are those used in the standard WHO-approved HI?); why are guinea pig RBCs included? - those are not analyzed later on in detail.

Response. Fowl erythrocytes, in particular those of chicken and turkey, are conventionally employed in HA assays because they are nucleated cells and display rapid sedimentation thereby greatly facilitating the visual readout of the assay. Erythrocytes from mammals are much more difficult to employ in HI assays because they do not readily sediment. Although guinea pig erythrocytes are difficult to employ in HI assays, they are occasionally used because these can agglutinate a somewhat broader range of viruses. We have included a section that clearly states why chicken and turkey erythrocytes are commonly employed in the HI assays. “*The glycan array was probed with various A/H3N2 viruses, representing distinct evolutionary time points and clades and having different abilities to agglutinate erythrocytes derived from species commonly used in HI assays (Supplementary Fig. 2). In this respect, chicken and turkey erythrocytes are widely employed in HA assays because they are nucleated, which unlike mammalian erythrocytes, sediment rapidly thereby greatly facilitating the visual readout of the assay⁷. They express α 2,3- and α 2,6-linked sialosides and can be agglutinated by avian as well as human influenza viruses. Other types of erythrocytes have been employed for HI assays²⁸ and in particular those of guinea pig erythrocytes have been proven to be useful because they exhibit a somewhat broader agglutination ability and can for example be employed to antigenically characterize H3N2 viruses of the 3C.2 clade which cannot be agglutinated turkey and chicken erythrocytes²⁹.*”

Remark. Page 7, Line 4+ - The comparison of the chicken and turkey RBC glycomics data were hard to compare as presented - maybe the key difference(s) that influence influenza binding can be highlighted - In Fig. 1 and elsewhere.

Response. **The color scheme for glycomic data has been changed to make comparison easier.** The following section describes the most important difference between the two cell types “*Chicken erythrocytes express substantial quantities of high mannose glycans (Supplementary Fig. 5) whereas turkey cells display almost none of these structures. The greater abundance of complex type glycans on turkey erythrocytes offers a possible rationale for the ability of the NL03 and NL09 A/H3N2 viruses to agglutinate unmodified or α 2,6-resialylated turkey erythrocytes, respectively.*”

To make it easier to understand Fig. 1B, the following text has been modified to aid clarity. “*The compounds are organized according to an increasing number of LacNAc moieties (indicated by different color coding) and include hybrid-type and core structures having none (black bars) or 1 LacNAc moiety (yellow bars) and complex N-glycans having variable numbers of LacNAc repeating units (2-4 LacNAc, green, blue and purple bars, respectively). High-mannose type N-glycans were also detected and their structures and abundance are shown in Supplementary Fig. 5.*”

Remark. Page 11, Line 2 and 4 - "human receptor" is not a useful term when it is used to mean alpha2,6-linked Sia, and the more complex extended glycans in the human respiratory tract. More more specific designations need to be given.

Response. The text has been modified and it is clearly state that human receptor refers to sialic acid alpha2,6-linked to galactose.

Remark. Page 11, Line 7 - what is the prototypic human receptor?

Response. As stated above, it refers to alpha2,6-linked to galactose. The text has been modified to avoid any ambiguity.

Remark. Page 12, Line 16 - this is Conclusions? Is this necessary if it basically repeats the abstract?

Response. The line has been removed and the first part of the conclusion section has been rewritten.

Remark. Page 14, Line 7 - describe the hCK cells.

Response. The following explanation has been provided. "(a humanized MDCK knock-out cell line, deficient of several 2,3-specific sialyltransferase enzymes and overexpressing human-type 2,6-sialylated receptors)³³". A reference has been added in which this cell line is described, ref. 33).

Reviewer 2.

Remark. The N-glycans synthesized all have the Asn residue at the reducing end. However, in the description of glycan array preparation, all N-glycans used for printing on the NHS glass surface have GlcNAc at the reducing end. This should be clarified and how the glycan reacted with the active ester on the glass surface should be stated. Another point is that the authors stated that airway epithelial cells contend extended LacNAc repeats on the cell surface.

Response. We thank the reviewer for the suggestion to clarify the text. All compounds have at the anomeric center a natural asparagine moiety. The α -amine of asparagine provides a reactive functionality for reaction with the succinate esters of the microarray slides. The text has been adjusted to make clear that all compounds containing an anomeric asparagine moiety and that its alpha amine is exploited for compounds immobilization on succinimide reactive microarray plates. "All glycans contain an anomeric asparagine moiety and its α -amine facilitated immobilization on amine reactive, NHS-activated glass slides. The quality of the printing was validated by probing the array with the lectins ECA, SNA and MAL1 as well as an anti-H3 antibody (Fig. 1A and Supplementary Fig. 1)."

Remarks. It would be better to clearly describe what type of extended structures and relative affinity for different viral hemagglutinins have been reported in the literature so the readers can have a better appreciation of the contributions from this group.

Response. The section dealing with glycan microarray development include the following line. "The glycans on the array resemble structures found in human respiratory tissue, which abundantly expresses N-glycans having multiple consecutive LacNAc repeating units that can be capped by a sialic acid^{25,26}." The conclusion section has been expanded and a paragraph has been included with a description what was known about recognition of extended structures (first two paragraphs).

Reviewer 3:

Remark. Although the focus is fowl erythrocytes, I would like some discussion on the relevance to human erythrocytes and glycans found on the respiratory track. Human erythrocytes are also used in the HIA assay. Most human glycans contain core fucose and multiple extended GlcNAc's do NOT occur (PLoS Pathog 9(3): e1003223. doi:10.1371/journal.ppat.1003223). Do the authors have human glycan data of relevance for human virus interactions that they could include? The data may also explain why certain sialylated inhibitors do not bind HA when expressed in CHO versus HEK cells where the nature of the sialylated linkages are different (J Immunol. 2020 Feb 15;204(4):1022-1034.

doi: 10.4049/jimmunol.1901145. Epub 2020 Jan 6.). Please expand the discussion to include these studies and the implication for human virus interactions.

Response. Fowl erythrocytes are commonly employed in the HA assay because they are small-nucleated cells that display rapid sedimentation thereby greatly facilitating the visual readout of the assay. Erythrocytes from mammals are difficult to employ in HA assays. It is important to note that influenza virus does not target erythrocytes for infection. Erythrocytes are employed to investigate the interaction of viruses with glycans. Human erythrocytes are rarely employed in HA assays. The text has been modified to clarify that fowl erythrocytes are commonly employed in HA assays because of convenience of use.

“The glycan array was probed with various A/H3N2 viruses, representing distinct evolutionary time points and clades and having different abilities to agglutinate erythrocytes derived from species commonly used in HI assays (Supplementary Fig. 2). In this respect, chicken and turkey erythrocytes are most commonly employed in HA assays because they are small and nucleated, which unlike mammalian erythrocytes, sediment rapid thereby greatly facilitating the visual readout of the assay⁷. They express α 2,3- and α 2,6-linked sialosides and can be agglutinated by bird as well as human influenza viruses. Other types of erythrocytes have been employed²⁸ and in particular those of guinea pig erythrocytes are useful because they exhibit a somewhat broader agglutination ability and can for example be employed to antigenically characterize H3N2 viruses of the 3C.2 clade which cannot be agglutinated turkey and chicken erythrocytes²⁹.”

Human respiratory tissue expresses *N*-glycans having multiple consecutive LacNAc repeating units that can be capped by sialic acids. The glycans on the microarray represent such structures. We do not propose that the *N*-glycans on the array and respiratory tract contain multiple extended GlcNAc's. They contain, however, epitopes composed of consecutive LacNAc repeating units (LacNAc is a disaccharide composed of Gal(1,4)GlcNAc). The referee is correct that *N*-glycans of the respiratory track can be modified by core fucose. However, core fucose is remote from where the virus binds the sialyl-LacNAc moiety of *N*-linked glycans and it is unlikely it influences recognition. The latter is supported by the modeling studies, which demonstrate that the reducing GlcNAc that can carry a core fucose is remote from where HA binds the glycan. The following statement has been added to clarify what types of *N*-glycans are expressed by human respiratory tissue: *“We have constructed a glycan array that is populated with biologically relevant bi-antennary *N*-glycans having different numbers of LacNAc repeating units in various structural configurations. They resemble structures found in human respiratory tissue, which abundantly expresses *N*-glycans having multiple consecutive LacNAc repeating units that can be capped by a sialic acid moiety^{25,26}.”* The modeling section and the conclusion reinforce that respiratory tissue expresses *N*-glycans modified by poly-LacNAc structures.

The referee is correct in stating that the nature of the host cell line can greatly affect the glycosylation of expressed proteins which in turn can affect biological properties. The requested statement and references have been added to the conclusion section.

Minor corrections

page 5 line 8, add anti-H3 antibody as this is what is shown in Fig 1A.

page 12, should be titled discussion not introduction.

Response. These corrections have been made.

Editorial remarks

A “Data Availability” section after the Methods section has been included.

A “Code Availability” section after the “Data Availability” section has been included. If the code can only be shared on request, please specify the reasons. For more information on our code sharing policy and requirements, please see:

<https://www.nature.com/nature-research/editorial-policies/reporting-standards#availability-of-computer-code>.

No database is available to deposit glycan microarray data, however, the raw data has been included in the supporting information (microarray source data).

The bar graphs contain individual data points.

Reviewer comments, second round –

Reviewer #1 (Remarks to the Author):

The revised manuscript addresses the main issues that I had about the previous version. The main advance here is a technical one about the assays that can be used to test for viral immunity using HI assays, and the changes make that clearer.

Reviewer #2 (Remarks to the Author):

The authors have clarified the questions and provided more detailed explanation of the result. The paper is acceptable for publication.

Reviewer #3 (Remarks to the Author):

All my concerns have been addressed.